# Are Dental Prophylaxis Protocols Safe for CAD-CAM Restorative Materials? Surface Characteristics and Fatigue Strength

Lucas Saldanha da Rosa [1], Luiza Freitas Brum Souza [1], Rafaela Oliveira Pilecco [2],
Thaís Andressa Cavalcante Kluch [1], Felipe Somavilla Binotto [1], Vitória Zanetti Henriques [1],
Cornelis Johannes Kleverlaan [3], Gabriel Kalil Rocha Pereira [1] and João Paulo Mendes Tribst [4,*]

[1]  Faculty of Dentistry, Federal University of Santa Maria (UFSM), Santa Maria 97105-900, Brazil;
    lucas.saldanha@acad.ufsm.br (L.S.d.R.); luizafbrum@hotmail.com (L.F.B.S.);
    thais.kluch@acad.ufsm.br (T.A.C.K.); somavillafelipe31@gmail.com (F.S.B.); vi.zanetti@hotmail.com (V.Z.H.);
    gabriel.pereira@ufsm.br (G.K.R.P.)
[2]  Department of Conservative Dentistry, Faculty of Dentistry, Federal University of Rio Grande do
    Sul (UFRGS), Porto Alegre 90035-004, Brazil; rafaela-pilecco@hotmail.com
[3]  Department of Dental Materials Science, Academic Centre for Dentistry Amsterdam (ACTA), University of
    Amsterdam and Vrije Universiteit Amsterdam, 1081LA Amsterdam, North Holland, The Netherlands;
    c.kleverlaan@acta.nl
[4]  Department of Reconstructive Oral Care, Academic Centre for Dentistry Amsterdam (ACTA), Universiteit
    van Amsterdam and Vrije Universiteit, 1081LA Amsterdam, North Holland, The Netherlands
*   Correspondence: j.p.mendes.tribst@acta.nl

**Abstract:** The surface of dental materials is exposed to various prophylaxis protocols during routine dental care. However, the impact of these protocols on the functional properties of the material's surface remains unclear. This study investigates the influence of different dental prophylaxis protocols on the surface properties and their effect on the mechanical performance of CAD-CAM restorative materials. Discs (Ø = 15 mm, thickness = 1.2 mm) were fabricated from resin composite (RC, Tetric CAD), leucite-reinforced (LEU, IPS Empress CAD), lithium disilicate (LD, IPS e.max CAD), and zirconia ceramics (ZIR, IPS e.max ZirCAD MT). The materials were subjected to six prophylactic treatments: untreated (CTRL), prophylactic paste fine (PPF), prophylactic paste coarse (PPC), pumice stone (PS), air abrasion with sodium bicarbonate jet (BJ), and ultrasonic scaling (US). Biaxial flexural fatigue tests, along with fractographic, roughness, and topographic analyses, were conducted. No significant changes in fatigue strength were observed for RC, LD, and ZIR under any prophylaxis protocols. However, LEU subjected to BJ treatment exhibited significantly reduced fatigue strength ($p = 0.004$), with a 22% strength reduction compared to the monotonic test and substantial surface alterations. Surface roughness analyses revealed increased roughness for RC treated with PPF, PPC, and PS compared to CTRL ($p < 0.05$), while LD exhibited decreased roughness following PPF, PS, and US treatments ($p < 0.05$). In ZIR, only the BJ protocol increased roughness ($p = 0.001$). In conclusion, dental prophylaxis protocols do not significantly affect the mechanical strength of RC, LD, and ZIR materials, thus allowing any protocol to be used for these materials. However, for LEU ceramics, the BJ protocol should be avoided due to its effect of reducing fatigue strength and damaging the surface.

**Keywords:** CAD/CAM; dental materials; prophylaxis paste; fatigue; prophylactic paste; surface roughness





## 1. Introduction

In recent years, the use of fixed dental prostheses fabricated using computer-aided design–computer-aided manufacturing (CAD-CAM) has become an increasingly common approach, particularly in the field of monolithic restorations. Extensive oral rehabilitations often require a complex combination of different restorative materials within the patient's mouth, as differences in stress incidence and aesthetic requirements between anterior

and posterior teeth lead to diverse material indications [1]. This scenario, combined with advances in biomaterial development, has resulted in the use of glass ceramics, such as leucite-reinforced and lithium disilicate-reinforced ceramics, as well as polycrystalline ceramics and even resin-based materials, to address the wide range of cases encountered in oral rehabilitation [2–4].

Attention should be given to the differences in microstructure, mechanical properties, and wear behavior among CAD-CAM materials [5]. For example, resin-based restorations have a lower elastic modulus compared to other restorative materials, which helps in stress distribution throughout the tooth structure [6]. However, this can also lead to increased abrasive wear of the restoration [7,8]. In contrast, glass ceramics exhibit greater stiffness, enhancing the wear resistance of the restoration, but they are more brittle and susceptible to subsurface damage, which can lead to fatigue failure [6,8,9]. Polycrystalline ceramics, while also prone to fatigue-related failure, are noted for their higher strength and stiffness compared to glass ceramics, though they are generally considered less aesthetically pleasing [6,10]. Consequently, these differences in material properties can inversely affect the susceptibility of the opposing tooth to abrasion, whereas resin would trigger less wear in comparison to ceramics [6,10].

Thus, the longevity of prosthetic restorations depends on factors beyond just material properties. Achieving success requires meticulous planning, precise execution, and the establishment of an effective maintenance protocol to ensure the health of surrounding tissues [11]. Patients commonly face challenges related to oral hygiene, such as biofilm accumulation and, in advanced cases, the development of calculus in hard-to-reach areas [12]. Additionally, the consumption of pigmented foods and beverages contributes to the staining of both teeth and restorations [13]. Dental prophylaxis protocols are therefore recommended to promote oral health by removing bacterial plaque, calculus, and external pigmentation stains [8,14]. These techniques vary based on the clinical scenario and professional preference, including the use of specific brushes or cups with prophylactic pastes or pumice stone for plaque removal, manual or ultrasonic scaling for calculus removal, and air polishing devices like the bicarbonate jet for extrinsic stain removal [8,14].

Despite the crucial role of periodic oral health maintenance using prophylaxis protocols, concerns have been raised about their impact on the surface properties of indirect materials [10,15–17]. Studies indicate that prophylaxis methods can significantly alter surface roughness, gloss, translucency, marginal integrity, and the presence of cracks in ceramic and resin-based materials [8,10,15–22]. Increased surface roughness is particularly critical, as it can enhance biofilm adhesion and maturation, leading to gum inflammation, patient discomfort, wear on opposing dentition, and color instability if surface finishing is inadequate [13,23–26]. Additionally, due to the brittleness of ceramic materials [4], surface deterioration, subsurface damage, and increased roughness can introduce defects and phase transformations, particularly in zirconia, impairing the mechanical properties of these materials through stress concentration [27,28]. The potential harmful effects of surface damage depend on the specific surface changes caused by each prophylaxis method and the material's response, which can vary due to differences in microstructure [5,9].

In vitro studies may predict the mechanical behavior of restorative materials by simulating oral cavity conditions, with fatigue tests providing more reliable data in this context than static tests, as oral function typically involves low loads over long periods [29,30]. As mentioned previously, there is a limited number of studies investigating the impact of various dental prophylaxis protocols on the surface properties of CAD-CAM materials. Despite that, to the authors' knowledge, there are no studies exploring the consequences of such protocols on the fatigue behavior of those indirect materials. Therefore, this study aimed to assess the effect of different dental prophylaxis protocols on the mechanical strength (monotonic and fatigue) and surface characteristics of various CAD-CAM materials. The null hypotheses assumed were as follows: [1] different prophylaxis protocols do not affect the fatigue flexural strength of different CAD-CAM materials, and [2] different prophylaxis protocols do not affect the surface characteristics of different CAD-CAM materials.

## 2. Materials and Methods

### 2.1. Materials and Study Design

The commercial name, manufacturer, composition, and batch number of the materials used in this study are listed in Table 1.

**Table 1.** Description of materials, commercial name, manufacturer, composition and batch number.

| Material | Commercial Name/Manufacturer | Composition | Batch Number |
|---|---|---|---|
| Resin composite | Tetric CAD HT A2/C14, Ivoclar AG, Schaan, Liechtenstein | Barium glass filler (64.0 wg%)<br>$SiO_2$ (7.1 wg%)<br>Dimethacrylates (28.4 wg%)<br>Additives & Pigments (0.5 wg%) | Z025G1 |
| Leucite | IPS Empress CAD LT A2/14, Ivoclar AG | $SiO_2$ (60.0–65.0 wg%)<br>$Al_2O_3$ (16.0–20.0 wg%)<br>$K_2O$ (10.0–14.0 wg%)<br>$Na_2O$ (3.5–6.5 wg%)<br>Other oxides (0.5–7.0 wg%)<br>Pigments (0.2–1.0 wg%) | V12349 |
| Lithium disilicate | IPS e.max CAD LT A2/16, Ivoclar AG | $SiO_2$ (57.0–80.0 wg%)<br>$Li_2O$ (11.0–19.0 wg%)<br>$K_2O$ (0.0–13.0 wg%)<br>Other oxides (0–8 wg%) | Z036HY |
| 4 mol% Yttrium-stabilized zirconia | IPS e.max ZirCAD MT A2, Ivoclar AG | $ZrO_2$ (86.0–93.5 wg%)<br>$Y_2O_3$ (>6.5%–≤8.0 wg%)<br>$HfO_2$ (≤5.0 wg%)<br>$Al_2O_3$ (≤1.0 wg%)<br>and other oxides (≤1.0 wg%) | X16331 |
| Prophylactic paste fine | Proxyt Fine, Ivoclar AG | $C_3H_8O_3$ (25–50 wg%)<br>$H_4Al_2O_9Si_2$ (10–25 wg%)<br>$TiO_2$ (<2.5 wg%) | Z040PG |
| Prophylactic paste coarse | Proxyt Coarse, Ivoclar AG | $C_3H_8O_3$ (25–50 wg%)<br>$TiO_2$ (<2.5 wg%)<br>$NaC_{12}H_{25}SO_4$ (<2.5 wg%)<br>NaF (<2.5 wg%) | Z03V1B |
| Pumice stone | Pomitec pumice stone, Iodontosul, Porto Alegre, RS, Brazil | $SiO_2$ | 12044 |
| Sodium bicarbonate | Airon sodium bicarbonate, Maquira Dental Group, Maringá, PR, Brazil | $NaHCO_3$<br>Colloidal silicic anhydride | 811221 |

This in vitro study involved six scenarios: five different prophylaxis protocols commonly used in clinical practice and one untreated control group, applied to four different CAD-CAM restorative materials, a resin composite, two glass ceramics, leucite and lithium disilicate and zirconia, and a polycrystalline ceramic. The groups underwent monotonic and fatigue biaxial flexural tests.

An illustration of the workflow for specimen manufacturing and prophylaxis protocols is shown in Figure 1.

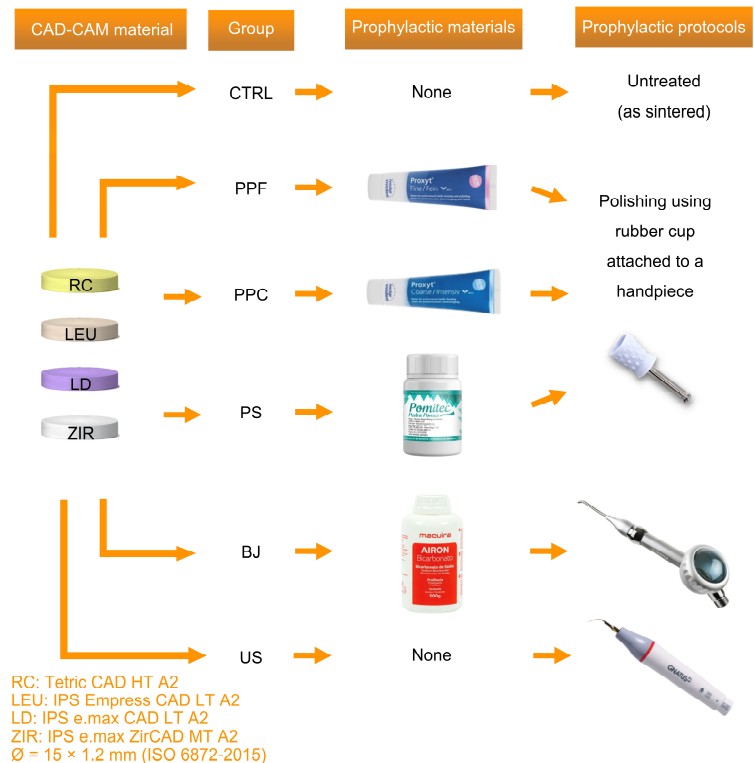

**Figure 1.** Flowchart illustrating the workflow for specimen manufacturing until the execution of the prophylaxis protocols. Materials labels: resin composite (RC), leucite (LEU), lithium disilicate (LD), and zirconia (ZIR). Group labels: untreated (CTRL), prophylactic paste fine (PPF), prophylactic paste coarse (PPC), pumice stone (PS), air abrasion with sodium bicarbonate jet (BJ), and ultrasonic scaling (US).

### 2.2. Specimen Preparation

The discs of the four CAD-CAM materials were manufactured according to ISO 6872-2015 standards [31]. The materials used were a resin composite (RC, Tetric CAD HT A2/C14, Ivoclar AG), a leucite-reinforced ceramic (LEU, IPS Empress CAD LT A2/14, Ivoclar AG), a lithium disilicate-reinforced ceramic (LD, IPS e.max CAD LT A2/C16, Ivoclar AG), and a zirconia ceramic (ZIR, IPS e.max 4YSZ ZirCAD MT A2, Ivoclar AG). For RC, LEU, and LD, two metal guides (Ø = 15 mm) were bonded to the blocks to guide shaping into cylinders using a polishing machine (EcoMet/AutoMet 250, Buehler, Lake Bluff, IL, USA) with a diamond disc (Dia-Grid Diamond Disks #120, Allied HighTech Products, Inc., Cerritos, CA, USA) and constant water cooling. Zirconia blocks were initially sectioned manually into square blocks (20 mm × 20 mm × 20 mm). Metallic guides (Ø = 18 mm) were then bonded with cyanoacrylate glue to the parallel surfaces of these blocks, which were subsequently shaped into cylinders using the same procedures, but with #600- and #1200-grit silicon carbide (SiC) papers instead of the diamond disc.

For all materials, the cylinders were sliced into discs (RC, LEU, and LD with 1.3 mm thick; ZIR with 1.5 mm thick) using a diamond disc on a precision cutting machine (Isomet 1000, Buehler, Lake Bluff, IL, USA), with constant water cooling. The discs underwent a final polishing procedure on both sides using #600-, #1200-grit (RC, LEU, LD, and ZIR) SiC papers to standardize the surfaces and remove any cutting irregularities. LEU and LD were crystallized in a furnace (Vacumat 6000 MP, Vita Zahnfabrik, Bad Säckingen, Germany), while ZIR was sintered (VITA Zyrcomat 6000 MS, Vita Zahnfabrik) according to the manufacturer's instructions, achieving final dimensions of Ø = 15 mm and thickness = 1.2 mm. The specimens were then randomly allocated into groups, as depicted in Figure 1.

### 2.3. Prophylaxis Protocols

For the control groups (CTRL), the specimens received no additional treatment. For the prophylactic paste fine (PPF) (Proxyt Fine, RDA 7, Ivoclar AG), prophylactic paste coarse (PPC) (Proxyt Coarse, RDA 83, Ivoclar AG), and pumice stone (PS) groups, a rubber cup was attached to a handpiece operating at 2000 rpm. One surface of each sample was treated with the respective prophylaxis protocols for 2 min with a standardized applied force of 3.9 N, measured using a weighing scale [10]. In the pumice stone (PS) group, the pumice powder (Pomitec pumice stone, Iodontosul, Porto Alegre, RS, Brazil) was mixed with distilled water at a ratio of 10 g:4 mL.

For the bicarbonate jet (BJ) group, a jet (Jet Sonic BP, Gnatus, Barretos, SP, Brazil) was used with maximum air pressure and volume, and 50% water flow, combined with sodium bicarbonate (Airon sodium bicarbonate, Maquira Dental Group, Maringá, PR, Brazil). The jet was positioned at a 90° angle and held 5 mm from the specimen surface for 20 s [20].

In the ultrasonic scaling (US) group, an ultrasonic tip (G4-G insert, Gnatus, Barretos, Brazil) was attached to an ultrasonic unit (Jet Sonic BP, Gnatus) with a pressure setting of 10%, and 50% water flow. The tip was positioned at a 15° angle, and horizontal sweeping movements with manual pressure were performed across the entire sample surface for 20 s [15,32]. A standardized applied force of 3.9 N was used, measured using a weighing scale [10].

### 2.4. Monotonic Biaxial Flexural Test

A biaxial flexural strength test was conducted using a monotonic approach (n = 15). The test was performed with a piston-on-three-balls setup according to ISO 6872-2015 standards [31] on a universal testing machine (DL-1000 Emic, Instron, São José dos Pinhais, PR, Brazil). The treated surface of the specimens was placed facing down upon three steel spheres (2.5 mm in diameter, spaced at 120° intervals to form a 10 mm diameter circle) submerged in distilled water. A cellophane tape (15 μm) was placed between the specimen and the spheres to reduce contact damage [33]. An adhesive tape (25 μm) was applied to the specimen's upper surface to prevent fragment scattering.

A load was applied at the center of the upper surface of the disc at a rate of 1 mm/min using a flat-tipped circular steel piston (Ø = 1.4 mm). The maximum fracture stress, measured in MPa, was calculated using the following equation:

$$\sigma_{BI} = \frac{-0.2837P(X - Y)}{b^2}$$

Here, $\sigma_{BI}$ is the maximum center tensile stress, in Megapascals; $P$ is the total load causing fracture (in Newtons); and $b$ is the specimen's thickness (in mm). The coefficients $X$ and $Y$ were determined as follows:

$$X = (1 + \nu)ln\left(\frac{r2}{r3}\right) + \left(\frac{1-\nu}{2}\right)\left(\frac{r2}{r3}\right)$$
$$Y = (1 + \nu)\left\lfloor 1 + ln\left(\frac{r1}{r3}\right)\right\rfloor + (1 - \nu)\left(\frac{r1}{r3}\right)$$

where $\nu$ is the Poisson's ratio ($\nu$ = 0.25; ISO 6872-2015), $r1$ is the radius of the support circle (in mm), $r2$ is the radius of the piston (in mm), and $r3$ is the radius of the specimen (in mm).

The monotonic flexural strength (in MPa) was recorded for subsequent statistical analysis.

### 2.5. Fatigue Biaxial Flexural Test

The cyclic fatigue test ($n$ = 15) was conducted using a mechanical testing machine (Instron Corporation; Norwood, MA, USA). The test geometry and setup were consistent with those used in the monotonic test. However, this test employed a cyclic fatigue method with a frequency of 20 Hz [34,35]. Using the mean flexural strength from the monotonic test as a baseline, the initial stress level for the cyclic fatigue test was set at 30%, with stress increments of 5% applied every 10,000 cycles. All samples were tested until fracture

occurred. The fatigue flexural strength (in MPa) and the number of cycles to failure (CFF) were recorded for statistical analysis.

### 2.6. Roughness Analysis

Before and after the prophylaxis protocols, the specimens ($n = 15$) from each group were subjected to micrometric roughness analysis using a contact stylus profilometer (Mitutoyo SJ-410, Kanagawa, Japan). Six measurements were taken for each specimen—three along each axis ($x$ and $y$)—with a cut-off ($\lambda$c) of 0.8 mm and a sampling length of 4 mm. The arithmetic mean was calculated for each specimen based on the parameters Ra (average surface roughness) and Rz (maximum peak-to-valley height), according to ISO 21920-2021 [36].

### 2.7. Fractographic and Topographic Analysis

After the fatigue test, all fractured specimens were examined under a stereomicroscope, and representative failures were selected for each group condition. These selected specimens were then analyzed using scanning electron microscopy (SEM; Vega3, Tescan, Brno, Czech Republic) at magnifications of $100\times$ and $5000\times$ to determine their fractographic features. Prior to this analysis, specimens underwent ultrasonic cleaning (1440 D Odontobras, Ribeirão Preto, Brazil) with 92% isopropyl alcohol for 5 min and were subsequently coated with a gold–palladium alloy.

Additionally, separate specimens for each condition were selected and examined using SEM (Vega3, Tescan) at magnifications of $100\times$ and $5000\times$ to assess the topographical characteristics of the ceramic surfaces after the different prophylaxis protocols. These ceramic specimens were also cleaned ultrasonically (1440 D Odontobras) with 92% isopropyl alcohol for 5 min and coated with a gold–palladium alloy before analysis.

### 2.8. Data Analysis

Since the monotonic strength and roughness data followed parametric and homoscedastic distributions, as confirmed by the Shapiro–Wilk and Levene's tests ($p > 0.05$), one-way ANOVA and Tukey's post hoc tests ($\alpha = 0.05$) were performed using SPSS version 21 (IBM Analytics, New York, NY, USA). For roughness analysis, comparisons between data before and after prophylaxis protocols within the same group were conducted using paired *t*-tests ($\alpha = 0.05$).

Fatigue flexural strength and cycles to failure (CFF) data were submitted to survival analysis by means of Kaplan–Meier and Mantel–Cox (Log-Rank) tests ($\alpha = 0.05$; SPSS version 21, IBM Analytics). Additionally, Weibull statistical analysis was performed on the fatigue flexural strength to characterize the material's reliability using SuperSMITH Weibull 4.0k-32 software (Wes Fulton, Torrance, CA, USA). Statistical differences were determined through maximum-likelihood estimation, with non-overlapping confidence intervals indicating significant differences in reliability.

Pearson correlation analysis was used to explore potential linear correlations between surface roughness means and both monotonic and fatigue strength. The Pearson correlation coefficient ranges from +1 (total positive linear correlation) to −1 (total negative linear correlation), with 0 indicating no linear correlation. The correlation intensity was categorized based on Cohen's classification [37]: 0.10 to 0.29 as weak, 0.30 to 0.49 as moderate, and 0.50 to 1.00 as strong. The coefficient of determination was also examined to assess how well the model fit the data.

Fractographic and topographic features were qualitatively analyzed.

## 3. Results

### 3.1. Resin Composite

Regarding monotonic strength, no significant differences were observed among the tested groups ($p > 0.05$). For fatigue data, no significant differences were found between any of the prophylaxis protocols and the untreated control group ($p > 0.05$) (Table 2 and Figure 2).

**Table 2.** Mean (in MPa), standard deviation (SD), and 95% confidence intervals (CIs) for the monotonic and fatigue biaxial flexural strength tests, and for number of cycles for failure (CFF) during fatigue testing.

| Group | | Monotonic (MPa) | | Fatigue (MPa) | | % Decrease | CFF (Counts) | |
|---|---|---|---|---|---|---|---|---|
| | | Mean ± SD * | 95% CI | Mean ± SD ** | 95% CI | | Mean ± SD ** | 95% CI |
| RC | CTRL | 212 ± 25 A | 198–226 | 172 ± 19 A | 161–182 | 18.9 | 94,648 ± 18,011 A | 84,674–104,622 |
| | PPF | 219 ± 38 A | 198–240 | 167 ± 17 A | 158–177 | 23.7 | 93,165 ± 16,953 A | 83,777–102,553 |
| | PPC | 214 ± 30 A | 197–230 | 167 ± 18 A | 157–177 | 21.9 | 94,275 ± 15,801 A | 85,525–103,026 |
| | PS | 232 ± 20 A | 220–243 | 167 ± 21 A | 156–179 | 28.0 | 92,760 ± 20,315 A | 81,510–104,011 |
| | BJ | 216 ± 21 A | 205–228 | 168 ± 13 A | 161–176 | 22.2 | 93,785 ± 12,093 A | 87,088–100,482 |
| | US | 214 ± 45 A | 189–239 | 169 ± 15 A | 161–177 | 21.0 | 93,753 ± 13,755 A | 86,136–101,371 |
| LEU | CTRL | 107 ± 8 A | 103–111 | 90 ± 11 AB | 83–96 | 15.9 | 113,786 ± 21,503 AB | 101,878–125,694 |
| | PPF | 101 ± 8 A | 97–106 | 84 ± 8 BC | 79–89 | 16.8 | 103,280 ± 16,564 BC | 94,107–112,453 |
| | PPC | 104 ± 14 A | 96–112 | 91 ± 10 A | 85–97 | 12.5 | 116,143 ± 20,295 A | 104,904–127,382 |
| | PS | 97 ± 6 A | 93–100 | 85 ± 11 B | 79–91 | 12.4 | 107,974 ± 16,338 B | 98,926–117,022 |
| | BJ | 103 ± 13 A | 96–110 | 80 ± 6 C | 76–83 | 22.3 | 93,785 ± 12,093 C | 87,089–100,482 |
| | US | 105 ± 10 A | 100–111 | 92 ± 14 A | 84–99 | 12.4 | 120,778 ± 20,923 A | 109,192–132,365 |
| LD | CTRL | 273 ± 55 BC | 242–303 | 194 ± 29 AB | 178–210 | 21.9 | 70,531 ± 19,182 AB | 59,909–81,154 |
| | PPF | 287 ± 50 ABC | 259–314 | 182 ± 50 B | 259–314 | 36.6 | 63,269 ± 18,736 B | 52,893–73,644 |
| | PPC | 260 ± 49 C | 233–287 | 173 ± 20 B | 162–184 | 33.5 | 55,386 ± 14,816 C | 47,181–63,591 |
| | PS | 323 ± 60 AB | 285–361 | 183 ± 27 B | 169–199 | 43.3 | 62,745 ± 18,664 B | 52,410–73,081 |
| | BJ | 294 ± 45 ABC | 269–319 | 209 ± 35 A | 190–229 | 28.9 | 81,401 ± 23,959 A | 68,133–94,669 |
| | US | 333 ± 62 A | 299–367 | 211 ± 33 A | 193–230 | 36.6 | 83,356 ± 22,357 A | 70,975–95,737 |
| ZIR | CTRL | 599 ± 108 A | 540–659 | 467 ± 68 AB | 430–505 | 22.0 | 93,459 ± 23,389 AB | 80,507–106,412 |
| | PPF | 582 ± 85 A | 535–630 | 459 ± 68 AB | 422–497 | 21.1 | 95,280 ± 23,371 AB | 82,337–108,222 |
| | PPC | 607 ± 93 A | 556–659 | 452 ± 63 AB | 417–486 | 25.5 | 97,496 ± 22,812 AB | 84,863–110,129 |
| | PS | 608 ± 144 A | 528–688 | 491 ± 57 A | 459–523 | 19.2 | 103,885 ± 19,300 A | 93,197–114,573 |
| | BJ | 561 ± 76 A | 519–603 | 430 ± 75 B | 389–471 | 23.3 | 84,060 ± 25,106 B | 70,157–97,964 |
| | US | 576 ± 91 A | 525–627 | 434 ± 80 B | 390–478 | 24.7 | 85,777 ± 26,866 B | 70,899–100,655 |

* Different uppercase letters indicate statistical differences depicted by one-way ANOVA and Tukey's post hoc tests ($\alpha$ = 0.05). ** Different uppercase letters indicate statistical differences depicted by Kaplan–Meier and Mantel–Cox post hoc tests ($\alpha$ = 0.05). Materials labels: resin composite (RC), leucite (LEU), lithium disilicate (LD), and zirconia (ZIR). Group labels: untreated (CTRL), prophylactic paste fine (PPF), prophylactic paste coarse (PPC), pumice stone (PS), air abrasion with sodium bicarbonate jet (BJ), and ultrasonic scaling (US).

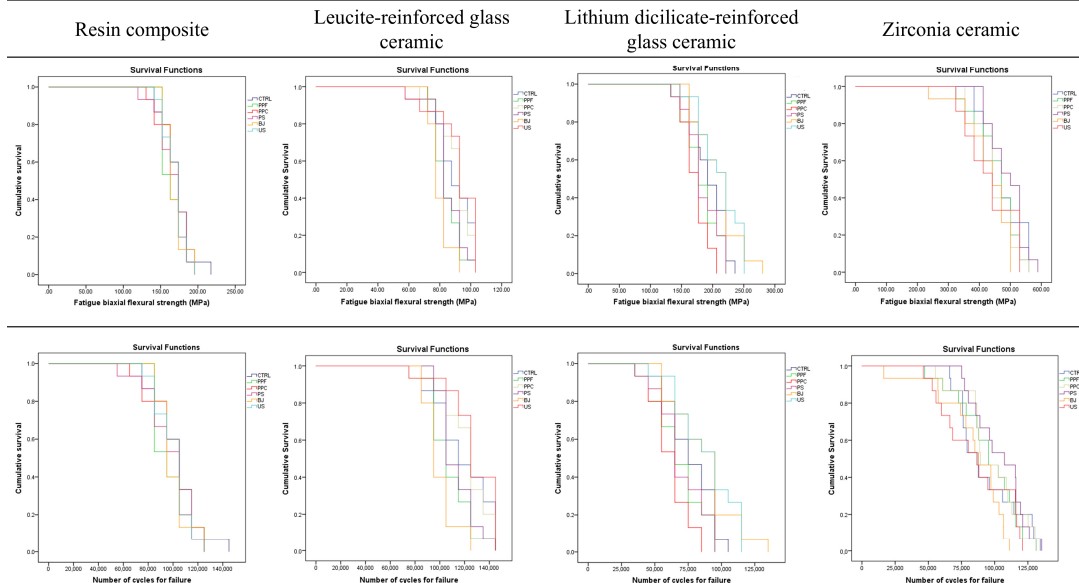

**Figure 2.** Survival plots for the different materials in the tested conditions showing the cumulative survival from 1.0 (100% survival) to 0.0 (100% failure) for fatigue biaxial flexural strength (MPa) and number of cycles for failure. Group labels: untreated (CTRL), prophylactic paste fine (PPF), prophylactic paste coarse (PPC), pumice stone (PS), air abrasion with sodium bicarbonate jet (BJ), and ultrasonic scaling (US).

In terms of surface roughness (Table 3), after treatment, only the BJ and US groups had similar Ra values to the CTRL group ($p > 0.05$). The PPC and PS groups showed increased Ra values compared to the CTRL ($p < 0.05$), and there was no significant difference between these two groups ($p > 0.05$). The PPF group also exhibited increased Ra values post-treatment ($p < 0.05$). Additionally, in the Rz parameter, only the US group showed no significant increase ($p > 0.05$) after treatment.

**Table 3.** Mean, standard deviation (SD), and 95% confidence intervals (CIs) for surface roughness parameters—Ra and Rz (in μm) pre- and post-prophylaxis protocols. The numeric difference between the Ra and Rz means pre- and post-procedure are shown by ΔRa and ΔRz (in μm).

| Group | | Ra (μm) | | Rz (μm) | | ΔRa | ΔRz |
|---|---|---|---|---|---|---|---|
| | | Pre | Post * | Pre | Post | | |
| | | Mean (SD) | Mean (SD) | Mean (SD) | Mean (SD) | | |
| RC | CTRL * | 0.04 (0.01) [A] | 0.04 (0.01) [C] | 0.37 (0.06) [AB] | 0.37 (0.06) [B] | - | - |
| | PPF | 0.04 (0.01) [Ab] | 0.07 (0.02) [Ba] | 0.37 (0.07) [ABb] | 0.71 (0.20) [Aa] | 0.03 | 0.34 |
| | PPC | 0.04 (0.01) [Ab] | 0.09 (0.01) [Aa] | 0.38 (0.06) [ABb] | 0.75 (0.11) [Aa] | 0.05 | 0.37 |
| | PS | 0.05 (0.01) [Ab] | 0.09 (0.01) [Aa] | 0.42 (0.04) [Ab] | 0.64 (0.11) [Aa] | 0.04 | 0.22 |
| | BJ | 0.04 (0.01) [Aa] | 0.05 (0.01) [Ca] | 0.33 (0.09) [Bb] | 0.45 (0.11) [Ba] | 0.01 | 0.12 |
| | US | 0.05 (0.01) [Aa] | 0.05 (0.03) [Ca] | 0.38 (0.06) [ABa] | 0.43 (0.10) [Ba] | 0.00 | 0.05 |
| LEU | CTRL * | 0.04 (0.02) [A] | 0.04 (0.02) [A] | 0.36 (0.10) [ABC] | 0.36 (0.10) [B] | - | - |
| | PPF | 0.03 (0.01) [Ab] | 0.05 (0.02) [Aa] | 0.35 (0.07) [ABCb] | 0.44 (0.07) [Ba] | 0.02 | 0.09 |
| | PPC | 0.04 (0.01) [Aa] | 0.05 (0.01) [Aa] | 0.39 (0.10) [ABa] | 0.40 (0.08) [Ba] | 0.01 | 0.01 |
| | PS | 0.04 (0.02) [Ab] | 0.06 (0.04) [Aa] | 0.30 (0.09) [BCb] | 0.72 (0.34) [Aa] | 0.02 | 0.42 |
| | BJ | 0.03 (0.01) [Aa] | 0.04 (0.05) [Aa] | 0.29 (0.08) [Ca] | 0.34 (0.11) [Ba] | 0.01 | 0.05 |
| | US | 0.05 (0.02) [Aa] | 0.04 (0.01) [Aa] | 0.41 (0.07) [Aa] | 0.49 (0.22) [Ba] | 0.01 | 0.08 |
| LD | CTRL * | 0.04 (0.01) [A] | 0.04 (0.01) [A] | 0.34 (0.05) [AB] | 0.34 (0.05) [AB] | - | - |
| | PPF | 0.04 (0.02) [Aa] | 0.03 (0.01) [Ba] | 0.38 (0.08) [Aa] | 0.31 (0.05) [BCb] | 0.01 | 0.07 |
| | PPC | 0.04 (0.01) [Aa] | 0.04 (0.01) [Aa] | 0.36 (0.07) [Aa] | 0.41 (0.08) [Aa] | 0.00 | 0.05 |
| | PS | 0.04 (0.01) [Aa] | 0.03 (0.01) [Bb] | 0.36 (0.07) [Aa] | 0.20 (0.08) [Db] | 0.01 | 0.16 |
| | BJ | 0.03 (0.01) [Ab] | 0.04 (0.01) [Aa] | 0.27 (0.09) [Bb] | 0.39 (0.12) [ABa] | 0.01 | 0.12 |
| | US | 0.03 (0.01) [Aa] | 0.03 (0.01) [Ba] | 0.19 (0.07) [Ca] | 0.23 (0.07) [CDa] | 0.00 | 0.04 |
| ZIR | CTRL * | 0.16 (0.02) [B] | 0.16 (0.02) [B] | 1.37 (0.30) [B] | 1.37 (0.30) [C] | - | - |
| | PPF | 0.22 (0.04) [Aa] | 0.22 (0.03) [ABa] | 2.01 (0.36) [Aa] | 1.93 (0.30) [ABa] | 0.00 | 0.08 |
| | PPC | 0.20 (0.03) [ABa] | 0.20 (0.04) [ABa] | 1.76 (0.33) [ABa] | 1.79 (0.41) [ABa] | 0.00 | 0.03 |
| | PS | 0.23 (0.04) [Aa] | 0.17 (0.05) [Bb] | 2.00 (0.32) [Aa] | 1.54 (0.47) [Bb] | 0.05 | 0.46 |
| | BJ | 0.24 (0.04) [Aa] | 0.26 (0.13) [Aa] | 2.16 (0.50) [Aa] | 2.23 (0.94) [Aa] | 0.02 | 0.07 |
| | US | 0.21 (0.34) [Aa] | 0.21 (0.05) [ABa] | 1.82 (0.40) [Aa] | 1.74 (0.38) [ABa] | 0.00 | 0.08 |

* CTRL post-treatments used values that were the same as the pretreatment values. Uppercase letters in each column represent statistical differences for roughness parameters among the groups in each moment (pre- or post-prophylaxis protocols) depicted by one-way ANOVA and Tukey's post hoc tests ($\alpha = 0.05$). Lowercase letters in each row represent statistical differences for roughness parameters within the same group pre- and post-prophylaxis protocols depicted by independent *t*-tests ($\alpha = 0.05$). Materials labels: resin composite (RC), leucite (LEU), lithium disilicate (LD), and zirconia (ZIR). Group labels: untreated (CTRL), prophylactic paste fine (PPF), prophylactic paste coarse (PPC), pumice stone (PS), air abrasion with sodium bicarbonate jet (BJ), and ultrasonic scaling (US).

The Pearson correlation analysis revealed a strong positive correlation between Ra and monotonic strength (Ra: r = 0.62) and a medium positive correlation between Rz and monotonic strength (Rz: r = 0.40). For fatigue strength, there was a strong negative correlation with Ra (r = −0.80) and Rz (r = −0.82). The Pearson coefficient of determination indicated that surface roughness (Ra: $R^2 = 0.64$; Rz: $R^2 = 0.68$) has a stronger correlation with fatigue strength compared to monotonic strength (Ra: $R^2 = 0.39$; Rz: $R^2 = 0.16$).

Fractographic analysis (Figure 3) revealed that fractures originated from surface defects in the region subjected to tensile stress concentration (treated surface) across all groups. Representative scanning electron microscopy (SEM) images of the material surfaces (Figure 4) displayed a consistent surface pattern among the different conditions,

where no visible modifications could be seen after the prophylaxis protocols compared to the CTRL condition.

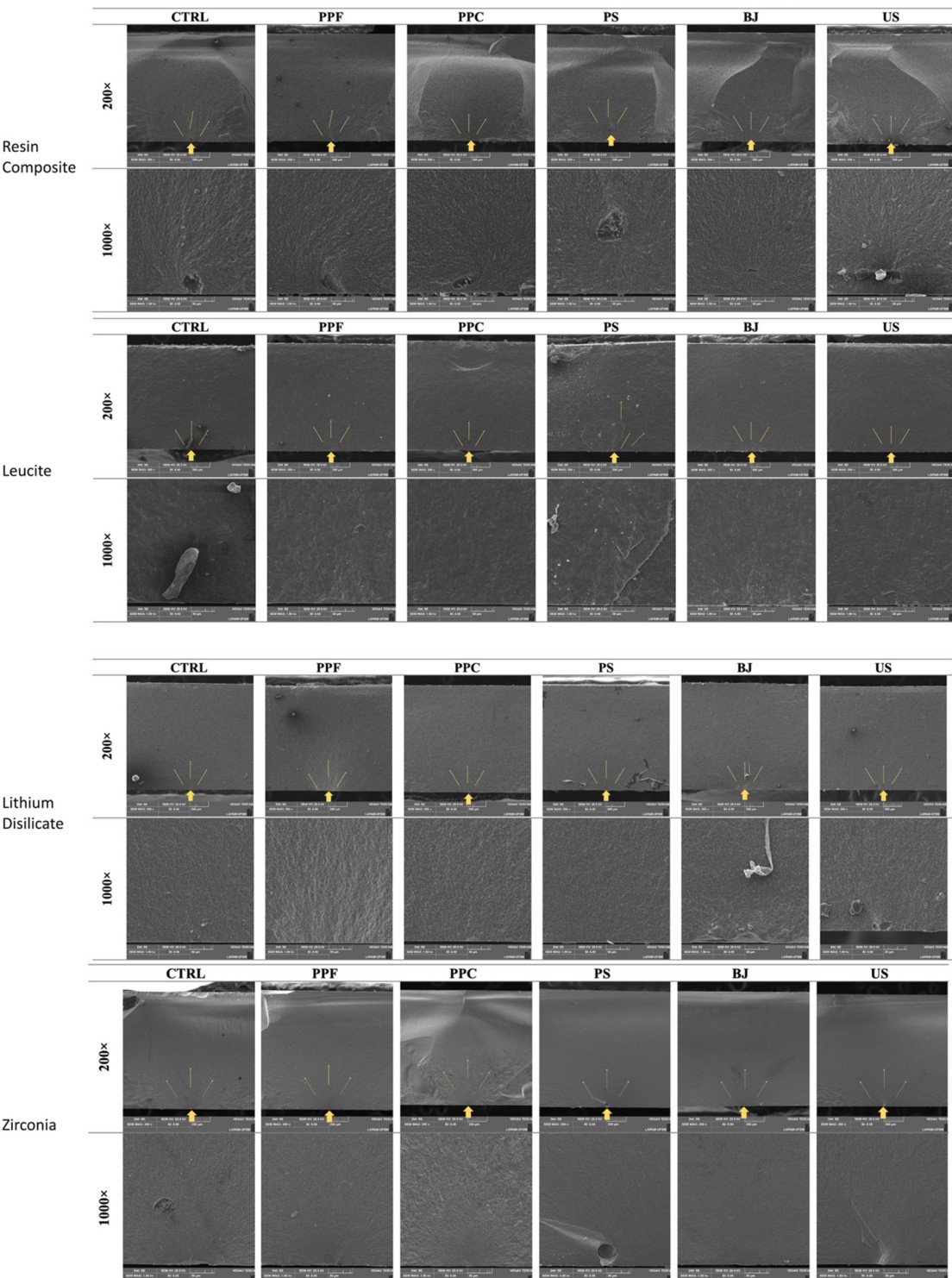

**Figure 3.** Representative SEM images of fractured resin composite, leucite, lithium disilicate, and zirconia surfaces of specimens subjected to fatigue tests. The origin of the fracture is pointed with the solid yellow arrows in regions that concentrated tensile stress during mechanical loading, and the thin yellow arrows indicate the crack propagation direction. Group labels: untreated (CTRL), prophylactic paste fine (PPF), prophylactic paste coarse (PPC), pumice stone (PS), air abrasion with sodium bicarbonate jet (BJ), and ultrasonic scaling (US).

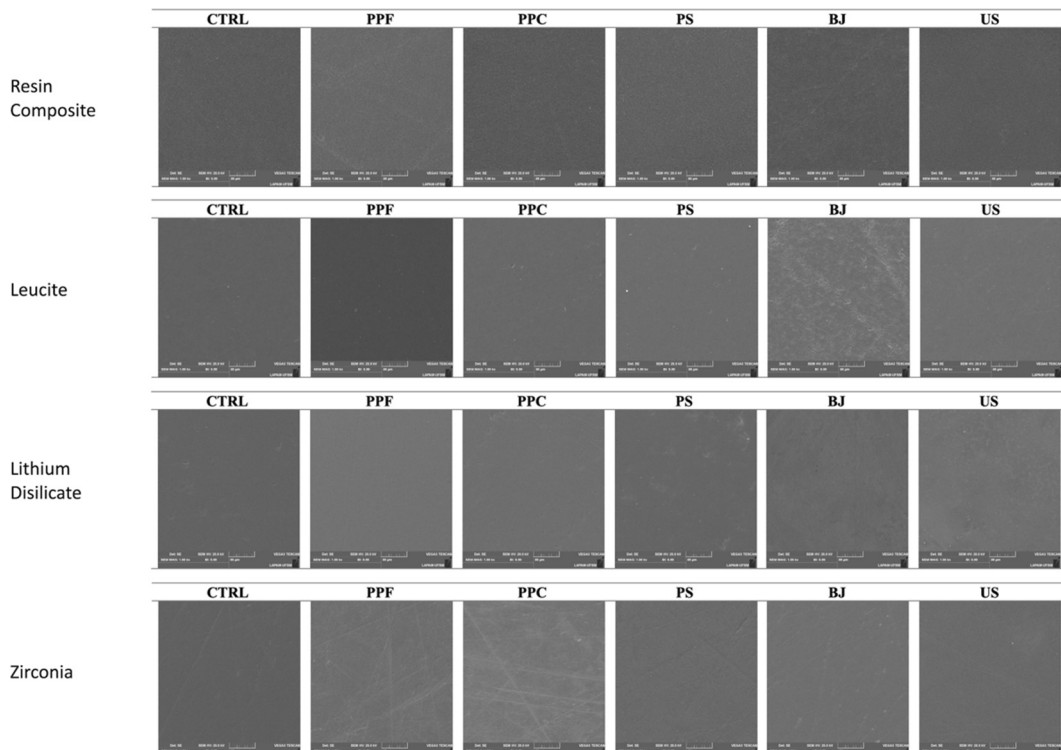

**Figure 4.** SEM micrographics illustrating surface topography of resin composite, leucite, lithium disilicate, and zirconia specimens after prophylaxis protocols at $1000\times$ magnification. Materials labels: resin composite (RC), leucite (LEU), lithium disilicate (LD), and zirconia (ZIR). Group labels: untreated (CTRL), prophylactic paste fine (PPF), prophylactic paste coarse (PPC), pumice stone (PS), air abrasion with sodium bicarbonate jet (BJ), and ultrasonic scaling (US). In RC and lithium disilicate, no visible modifications could be seen after the prophylaxis protocols compared to the CTRL condition. In leucite, it can be seen that the bicarbonate jet (BJ) promoted perceptible surface alterations with the partial removal of the ceramic surface content in some areas. In zirconia, PPF, and especially PPC, treatment promoted a more scratched surface.

### 3.2. Leucite-Reinforced Ceramic

Regarding monotonic strength (Table 2 and Figure 2), no significant differences were observed among the tested groups ($p > 0.05$). In terms of fatigue behavior, only the BJ group showed a significant difference ($p = 0.004$), with no other significant differences found between the treated groups and the untreated control. Additionally, the PPC and US groups performed better than the PPF, PS, and BJ groups. The PPF group was similar to both the PS and BJ groups.

For surface roughness (Table 3), none of the treatments significantly altered the Ra values compared to the CTRL group ($p > 0.05$). However, in terms of Rz, the PS group was the only one to present higher values than CTRL ($p = 0.000$). The PPF and PS treatments led to an increase in both Ra and Rz ($p < 0.05$).

The Pearson correlation analysis showed a strong negative correlation between monotonic strength and surface roughness (Ra: r = −0.87; Rz: r = −0.79), whereas the correlation with fatigue strength was weak, with Rz showing no correlation (Ra: r = −0.14; Rz: r = 0.00). The Pearson coefficient of determination indicated that surface roughness (Ra: $R^2 = 0.75$; Rz: $R^2 = 0.63$) is a better predictor of monotonic strength compared to fatigue strength (Ra: $R^2 = 0.02$; Rz: $R^2 = 0.00$).

Fractographic analysis (Figure 3) revealed that fractures originated from surface defects in the region subjected to tensile stress concentration (treated surface) across all groups. The representative scanning electron microscopy (SEM) images of the material surfaces (Figure 4) displayed a consistent surface pattern among the different conditions,

except for the BJ group, which exhibited perceptible surface alterations compared to the untreated condition.

### 3.3. Lithium Disilicate-Reinforced Ceramic

In the monotonic tests, the PS and US groups exhibited higher fracture strength values compared to the PPC group ($p < 0.05$). Additionally, the US group demonstrated better performance than the CTRL group. For fatigue behavior, with the exception of PPC, no significant differences were observed between the treated groups and the CTRL group ($p > 0.05$) (Table 2 and Figure 2). Furthermore, the US and BJ groups showed better fatigue performance compared to the PPF, PPC, and PS groups ($p < 0.05$).

In the roughness analysis (Table 3), the Ra values for the PPC and BJ groups did not differ significantly from the CTRL group ($p > 0.05$). However, regarding Rz, the PS and US groups had statistically lower roughness values than the CTRL group ($p < 0.05$). The treatments altered both Ra and Rz, with the BJ group increasing these values ($p < 0.05$) and the PS group decreasing them ($p < 0.05$). The Rz parameter was also decreased in the PPF group ($p = 0.000$).

The Pearson correlation analysis revealed a strong negative correlation for monotonic strength with Ra (r = $-0.75$) and Rz (r = $-0.87$), whereas the correlation with fatigue strength was weak, with Ra showing no correlation (Ra: r = $0.00$; Rz: r = $-0.14$). The Pearson coefficient of determination indicated that monotonic strength (Ra: $R^2 = 0.56$; Rz: $R^2 = 0.75$) is better supported by roughness than fatigue strength (Ra: $R^2 = 0.00$; Rz: $R^2 = 0.02$).

Fractographic analysis (Figure 3) showed that fractures originated from surface defects in the region subjected to tensile stress concentration (treated surface) across all groups. Representative scanning electron microscopy (SEM) images of the material surfaces (Figure 4) displayed a consistent surface pattern among the different conditions, even after the application of prophylaxis protocols.

### 3.4. Zirconia Ceramic

Regarding monotonic strength, no significant differences were observed among the tested groups ($p > 0.05$). In the fatigue analysis, the PS group demonstrated superior fatigue behavior compared to both the BJ ($p = 0.02$) and US ($p = 0.03$) groups (Table 2 and Figure 2). For this ceramic, no significant differences in surface roughness (Ra and Rz parameters) were observed among the treated groups and the untreated group ($p > 0.05$), except for the BJ group, which exhibited higher surface roughness compared to the CTRL ($p = 0.001$) and PS groups ($p < 0.01$) (Table 3). Notably, only the PS group showed a decrease in surface roughness after prophylaxis protocols (Ra and Rz, $p = 0.01$) (Table 3).

The Pearson correlation analysis revealed a strong negative correlation between monotonic strength and roughness (Ra: r = $-0.85$; Rz: r = $-0.76$), as well as between fatigue strength and roughness (Ra: r = $-0.83$; Rz: r = $-0.72$). The Pearson coefficient of determination indicated that monotonic strength (Ra: $R^2 = 0.73$; Rz: $R^2 = 0.58$) was more strongly supported by roughness than fatigue strength (Ra: $R^2 = 0.62$; Rz: $R^2 = 0.48$).

Fractographic analysis (Figure 3) showed that fractures originated from surface defects in the region subjected to tensile stress concentration (treated surface) in all groups. The representative scanning electron microscopy (SEM) images of the material surface (Figure 4) displayed a consistent surface pattern across different conditions, with no visible modifications following the application of prophylaxis protocols.

## 4. Discussion

According to the findings of this study, dental prophylaxis protocols involving fine or coarse prophylactic paste, pumice stone, sodium bicarbonate jet, and ultrasonic scaling did not affect the fatigue flexural strength of CAD-CAM resin composite, lithium disilicate-reinforced ceramic, and zirconia ceramic compared to an untreated condition. Only the sodium bicarbonate jet treatment impaired the fatigue flexural strength of the CAD-CAM

leucite-reinforced ceramic. This leads to the partial rejection of the first hypothesis. However, increased surface roughness was observed following prophylaxis protocols compared to the untreated condition for all materials except leucite-reinforced ceramic. This results in the partial rejection of the second null hypothesis.

Regarding mechanical aspects, dental prophylaxis protocols did not significantly affect fatigue flexural strength in resin composite, lithium disilicate-reinforced ceramic, or zirconia ceramic when compared to the control condition. Resin composites consist of a complex organic polymeric matrix reinforced by inorganic filler particles [38]. Due to their nature, these materials can withstand high loads before failure, as resin-based materials exhibit a lower elastic modulus and pronounced resilient behavior [39]. Lithium disilicate-reinforced ceramics are known for their satisfactory mechanical and optical properties [40]. Compared to feldspathic and leucite-reinforced ceramics, lithium disilicate demonstrates increased fracture resistance [4,41]. Zirconia, a polycrystalline ceramic composed of oxides and grains without a glassy matrix [42], possesses a dense crystal network that enhances fracture toughness, resulting in high flexural strength [43]. Given these microstructural characteristics, no significant topographic changes were noted in these materials (Figure 4). Despite some roughness alterations after prophylaxis protocols, these changes were not substantial enough to compromise the mechanical behavior of resin composite, lithium disilicate, and zirconia, resulting in similar flexural strength to the untreated condition.

The only material negatively impacted by dental prophylaxis protocols was the leucite-reinforced ceramic. Among the tested materials, leucite-reinforced ceramic has the lowest fracture strength [3] despite its satisfactory aesthetics [44]. The leucite glassy matrix is reinforced with leucite crystals, which are lamina-like and range from 1 to 5 μm in length. This differs from lithium disilicate-reinforced ceramics, which are filled with needle-like lithium disilicate crystals, 1 μm long and 0.4 μm wide [45–47]. These needle-like crystals can act as a crack growth-stopping mechanism [4]. Although no significant roughness variation (Ra) was observed among the different dental prophylaxis protocols (Table 3), noticeable topographic alterations occurred with the use of the bicarbonate jet (Figure 4). This is consistent with the higher glass matrix content in leucite-reinforced ceramics, which was twice that of lithium disilicate-reinforced ceramics, according to the manufacturer's information. Additionally, leucite crystals lacked a strong interlocking effect and therefore failed to stop crack growth initiated by the impact of sodium bicarbonate particles, and although they had a relatively low hardness on Moh's scale (2.0), they were used under high pressure (6 bar) in this prophylactic protocol [3,4,45,48]. This was reflected in the fatigue strength results, which were negatively affected by the bicarbonate jet protocol (Table 2).

For CAD-CAM materials, surface roughness was altered by dental prophylaxis protocols, except in leucite-reinforced ceramic. In the resin composite, the fine and coarse prophylactic pastes and pumice stone protocols increased roughness, resulting in significantly rougher surfaces than the control group. These prophylaxis protocols use products applied with a rubber cup, which could account for the observed roughness results. However, these increased roughness values did not affect fatigue strength due to the resin composite's inherent physical properties [39]. In lithium disilicate-reinforced ceramics, while the bicarbonate jet slightly increased roughness, this effect was not noticeable in SEM images due to the well-packed crystals leaving only small portions of the glassy matrix exposed [3,4]. Conversely, the pumice stone protocol significantly reduced roughness due to its polishing effect, with pronounced differences compared to the control group. Nevertheless, similar to the resin composite material, these roughness alterations did not significantly affect the fatigue resistance of the lithium disilicate-reinforced ceramic.

The pumice stone protocol also reduced roughness in zirconia ceramic, indicating its high polishing properties. This reduction in roughness was observed across all tested materials except leucite-reinforced ceramic. Pumice stone is known for its high hardness (7 on the Mohs scale) [49] and notable relative dentin abrasivity (RDA) of at least 143 [50]. However, the bicarbonate jet significantly increased roughness compared to the untreated

group, likely due to the large sodium bicarbonate particles (up to 250 μm) [51], which can cause substantial damage such as erosive wear and deep defects [52]. In materials with increased mechanical strength, such as zirconia, these effects do not significantly impact mechanical fatigue behavior compared to more sensitive materials like leucite-reinforced ceramics.

Other dental prophylaxis protocols generally did not cause significant roughness alterations compared to the untreated group. Ultrasonic scaling, for instance, has been reported to cause surface scratching without significant changes in Ra values [17]. In this study, it led to slight roughness alterations in lithium disilicate-reinforced ceramic, which were significant compared to the control but did not impair fatigue strength. The fine and coarse prophylactic pastes used in this study consist of kaolin ($H_4Al_2O_9Si_2$) (2 on the Mohs scale) and titanium dioxide ($TiO_2$) (5.5 on the Mohs scale), respectively [49]. This may explain why these treatments did not alter the zirconia's surface roughness. The pastes also have RDAs of 7 for fine paste and 83 for coarse paste, indicating low and high abrasive content, respectively [53]. While both pastes increased roughness in resin composite, this did not affect fatigue behavior due to the material's resilient properties [39]. For lithium disilicate-reinforced ceramic, the coarse prophylactic paste did not alter roughness, whereas the fine paste had a polishing effect with significantly lower Ra values compared to the control group. However, these changes did not significantly affect fatigue strength due to the material's high-crystalline structure [3,4].

Reduced surface roughness is generally associated with positive effects on mechanical strength [54]. However, in this study, Pearson's linear correlation coefficient showed differing trends depending on the substrate. For materials without a glassy matrix (resin composite and zirconia), there was a high negative correlation, indicating that increased roughness decreases fatigue strength. For glass ceramics (leucite-reinforced and lithium disilicate-reinforced), there was a weak correlation between roughness and fatigue strength. Fraga et al. [54] (2017) observed lower fatigue strength in milled leucite-reinforced and zirconia ceramics, while lithium disilicate-reinforced ceramics did not show this effect, aligning with our study for zirconia and lithium disilicate. The weak correlation in leucite-reinforced ceramic suggests that other factors may also influence the results.

Surface roughness of restorations plays a crucial role in treatment success, influencing patient comfort, abrasiveness on opposing dentition, and color stability [13,23–25]. Some dental prophylaxis protocols significantly increased roughness in analyzed substrates. However, the roughness differences ranged from 0.00 to 0.05 μm, well below the critical limit of 0.2 μm for microbial plaque accumulation [55,56], which could affect biofilm adhesion and maturation [26]. In some cases, protocols like pumice stone reduced surface roughness, potentially mitigating these issues. Although no polishing was conducted post-prophylaxis, using specific polishing kits for each material should be beneficial in reducing surface roughness and enhancing restoration longevity [57].

Finally, it is important to acknowledge the study's limitations, including the absence of anatomical restorations and unexamined effects after bonding to a substrate. The testing setup used only axial loads during mechanical tests, not accounting for sliding or lateral forces that might occur in clinical settings. The load was applied opposite the prophylaxis treatments to isolate the protocol factor, differing from a clinical scenario. The decision not to use post-sintering polishing protocols or glaze applications was intentional to isolate the influence of prophylaxis protocols as a single factor. It is important to note that various prophylactic pastes and air-polishing powders can affect mechanical and surface properties differently due to variations in composition and recommended protocols. Thus, those results are specific to the materials used in this study.

## 5. Conclusions

Dental prophylaxis protocols that include prophylactic paste fine, prophylactic paste coarse, pumice stone, air abrasion with sodium bicarbonate jet, and ultrasonic scaling do not adversely affect the mechanical strength of CAD-CAM resin composite, leucite-

reinforced, lithium disilicate-reinforced, and zirconia ceramics. This is excepted for leucite-reinforced ceramic, in which sodium bicarbonate jet negatively impacted fatigue strength. Additionally, these protocols had only minor effects on surface characteristics and were not able to make clinically significant changes.

**Author Contributions:** Conceptualization, L.S.d.R., L.F.B.S., R.O.P., J.P.M.T., C.J.K., and G.K.R.P.; methodology, L.S.d.R., L.F.B.S., R.O.P., J.P.M.T., C.J.K., and G.K.R.P.; software, L.S.d.R., L.F.B.S., R.O.P., and G.K.R.P.; validation, L.S.d.R., L.F.B.S., R.O.P., T.A.C.K., J.P.M.T., C.J.K., and G.K.R.P.; formal analysis, L.S.d.R., L.F.B.S., R.O.P., T.A.C.K., F.S.B., V.Z.H., J.P.M.T., C.J.K., and G.K.R.P.; investigation, L.S.d.R., L.F.B.S., R.O.P., T.A.C.K., F.S.B., V.Z.H., J.P.M.T., C.J.K., and G.K.R.P.; resources, J.P.M.T., C.J.K., and G.K.R.P.; data curation, L.S.d.R., L.F.B.S., R.O.P., T.A.C.K., F.S.B., V.Z.H., and G.K.R.P.; writing—original draft preparation, L.S.d.R., T.A.C.K., and G.K.R.P.; writing—review and editing, L.S.d.R., L.F.B.S., R.O.P., T.A.C.K., F.S.B., V.Z.H., J.P.M.T., C.J.K., and G.K.R.P.; visualization, L.S.d.R., L.F.B.S., R.O.P., T.A.C.K., F.S.B., V.Z.H., J.P.M.T., C.J.K., and G.K.R.P.; supervision, L.S.d.R., L.F.B.S., R.O.P., J.P.M.T., C.J.K., and G.K.R.P.; project administration, J.P.M.T., C.J.K., and G.K.R.P.; funding acquisition, G.K.R.P. All authors have read and agreed to the published version of the manuscript.

**Funding:** This research was funded by the Brazilian Federal Agency for Coordination of Improvement of Higher Education Personnel (CAPES) (L.S.D.R. Doctorate's scholarship; Finance Code 001), by the Brazilian National Council for Scientific and Technological Development (CNPq) (R.O.P. doctorate scholarships, #140118/2022-5; L.F.B.S. doctorate scholarships, #162322/2022-4, and G.K.R.P. research Grant, #304665/2022-3) and by Research Foundation of the Rio Grande do Sul (FAPERGS) (Grants #24/2551-0001408-0).

**Institutional Review Board Statement:** Not applicable.

**Informed Consent Statement:** Not applicable.

**Data Availability Statement:** Data are available upon reasonable request for the first author.

**Acknowledgments:** The authors have no relevant financial or non-financial interests to disclose. We especially thank Ivoclar AG for donating some materials and emphasize that those institutions had no role in the study design, data collection or analysis, decision to publish, or manuscript preparation.

**Conflicts of Interest:** The authors declare no conflicts of interest.

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
