# Peer review of "Are Dental Prophylaxis Protocols Safe for CAD-CAM Restorative Materials? Surface Characteristics and Fatigue Strength"

_coatings, doi:10.3390/coatings14121510_

Round 1

Reviewer 1 Report

Comments and Suggestions for Authors

Dear Editor,

coatings-3337295,

The study presented in the manuscript examines the impact of various dental prophylaxis protocols on CAD-CAM restorative materials, with a focus on their surface characteristics and mechanical performance. While the research provides valuable information about the short-term effects of prophylactic treatments, there are several aspects that could be enhanced or require additional explanation to improve the paper's credibility and applicability.

1-    Regarding the fractographic and topographic analysis, the discussion on how bicarbonate jet treatment affects LEU ceramics' surface could be more comprehensive. Specifically, elaborating on the underlying mechanisms, such as the role of material composition and microstructure in the observed damage, would offer a more in-depth understanding of the results.

2-    In terms of methodology and specimen preparation, the rationale behind the chosen specimen dimensions (Ø 15 mm, thickness 1.2 mm) should be better supported by citing industry standards or previous studies. Moreover, including a brief explanation for selecting specific CAD-CAM materials (e.g., resin composite, leucite-reinforced ceramic) for testing would provide more context and help readers grasp the reasoning behind these choices.

3-    Figure 2 is not clear enough.

4-    The surface roughness analysis section, which examines changes in Ra and Rz, would benefit from a more detailed explanation, particularly regarding the implications for material performance. Further clarification on how these alterations in roughness affect longevity and clinical use of the materials would enhance the overall significance of the findings.

5-    The conclusion section is too short.

Author Response

The study presented in the manuscript examines the impact of various dental prophylaxis protocols on CAD-CAM restorative materials, with a focus on their surface characteristics and mechanical performance. While the research provides valuable information about the short-term effects of prophylactic treatments, there are several aspects that could be enhanced or require additional explanation to improve the paper's credibility and applicability.

Answer: Thank you for your comments. We are glad for the opportunity to consider your observations and improve our research to meet the journal high standards. We hope that all your comments were answered and, please, contact us for any further clarifications.

1- Regarding the fractographic and topographic analysis, the discussion on how bicarbonate jet treatment affects LEU ceramics' surface could be more comprehensive. Specifically, elaborating on the underlying mechanisms, such as the role of material composition and microstructure in the observed damage, would offer a more in-depth understanding of the results.

Answer: Thank you for your comment. We reviewed the manuscript and hopefully improved such section, as you can see the leucite’s microstructure is addressed in lines 378 to 383. There we discuss about the high glass matrix content in this ceramic and how the lack of long reinforcement crystals, such as in lithium disilicate ceramics, make it easier to be damaged by bicarbonate jet particles, that impact the material in a high pressure (6 bar).

2- In terms of methodology and specimen preparation, the rationale behind the chosen specimen dimensions (Ø 15 mm, thickness 1.2 mm) should be better supported by citing industry standards or previous studies. Moreover, including a brief explanation for selecting specific CAD-CAM materials (e.g., resin composite, leucite-reinforced ceramic) for testing would provide more context and help readers grasp the reasoning behind these choices.

Answer: Thank you for your comment. The specimen preparation was based in the ISO 6872-2015 standards, as shown in lines 116 and 117. We also reviewed the explanation for each material in the Materials and study design section.

3- Figure 2 is not clear enough.

Answer: Thank you for your comment. We enhanced the figure caption.

4- The surface roughness analysis section, which examines changes in Ra and Rz, would benefit from a more detailed explanation, particularly regarding the implications for material performance. Further clarification on how these alterations in roughness affect longevity and clinical use of the materials would enhance the overall significance of the findings.

Answer: Thank you for mentioning that. Although lower roughness being associated to better mechanical performance, we performed a correlation analysis between roughness and mechanical performance and we did not find strong correlations. This indicates that other factors were contributing to our findings as the materials microstructure, which was explored along the Discussion section. Furthermore, we discuss the impact of restoration’s surface roughness in lines 439 to 444.

5- The conclusion section is too short.

Answer: Thank you for pointing this. We enhanced the conclusion to better reflect our findings.

Reviewer 2 Report

Comments and Suggestions for Authors

Nice article , good written. Small minor improvements needed:

Abstract

It would be good to provide specific numerical values ​​of the obtained results, e.g. did the comp strength reduce from 150 MPa to 120 MPa or 20%?

Introduction

Please think about what your null hypothesis is.

https://en.wikipedia.org/wiki/Null_hypothesis

Materials and methods

Line 128

Metallic guides (Ø= 18 mm) were then bonded to the parallel surfaces of these blocks- how were they attached with what glue?

 Resutls

Figure 2 under the figure there are colored lines with abbreviations, it would be good to expand them so that the figure taken out of the text was understandable. Thanks

The same for other figures and tables. Please add a shortened text under each of them. It is easier to read.

I like the discussion. Good luck with your further research!

Author Response

Nice article, good written. Small minor improvements needed:

Answer: Thank you for your words, time and considerations. We hope to properly improve our work according to your comments.

Abstract

It would be good to provide specific numerical values of the obtained results, e.g. did the comp strength reduce from 150 MPa to 120 MPa or 20%?

Answer: Thank you for your comment. We adjusted the abstract to reflect the strength reduction you mentioned.

Introduction

Please think about what your null hypothesis is. https://en.wikipedia.org/wiki/Null_hypothesis

Answer: Thank you for your reflection. We checked the material you provided and we understand that the presented null hypotheses are aligned to the consulted concepts. If you disagree and want to recommend any change, we can adjust as desired.

Materials and methods

Line 128

Metallic guides (Ø= 18 mm) were then bonded to the parallel surfaces of these blocks- how were they attached with what glue?

Answer: Thank you for pointing this. A cyanoacrylate glue was used. We added this information to the text.

Resutls

Figure 2 under the figure there are colored lines with abbreviations, it would be good to expand them so that the figure taken out of the text was understandable. Thanks

Answer: Thank you for your comment. We added information to the figure for better understanding.

The same for other figures and tables. Please add a shortened text under each of them. It is easier to read.

Answer: Thank you for pointing this. We enhanced the figures and tables captions for better reading.

I like the discussion. Good luck with your further research!

Answer: Thank you